# Conventional vs. Microwave-Assisted Hydrodistillation: Influence on the Chemistry of Sea Fennel Essential Oil and Its By-Products

**DOI:** 10.3390/plants12071466

**Published:** 2023-03-27

**Authors:** Olivera Politeo, Marijana Popović, Maja Veršić Bratinčević, Petra Koceić, Tonka Ninčević Runjić, Ivana Generalić Mekinić

**Affiliations:** 1Department of Biochemistry, Faculty of Chemistry and Technology, University of Split, R. Boškovića 35, HR-21000 Split, Croatia; 2Department of Applied Science, Institute for Adriatic Crops and Karst Reclamation, Put Duilova 11, HR-21000 Split, Croatia; 3Department of Food Technology and Biotechnology, Faculty of Chemistry and Technology, University of Split, R. Boškovića 35, HR-21000 Split, Croatia; 4Department of Plant Science, Institute for Adriatic Crops and Karst Reclamation, Put Duilova 11, HR-21000 Split, Croatia

**Keywords:** sea fennel, essential oils, hydrolates, residual wastewater, HS-SPME, GC-MS, HPLC

## Abstract

The main objectives of this study were to investigate the effects of the applied essential oil (EO) isolation method, conventional hydro-distillation (HD), and microwave-assisted hydro-distillation (MHD) on the chemical profile of sea fennel (*Crithmum maritimum* L.) essential oil and to investigate the main constituents present in the liquid by-products of EOs isolation (hydrolate and residual wastewater). Headspace-solid phase microextraction (*HS-SPME*) was used to isolate hydrolate components, while gas chromatography coupled with mass spectrometry (GC-MS) was used to detect and analyse the chemical constituents of the essential oils and hydrolates. The phenolic composition of the wastewater extracts was analysed by high performance liquid chromatography (HPLC). The EO obtained by MHD had a higher yield of limonene and sabinene. The chemical composition of the hydrolates differed from the EO compositions. The content of terpinen-4-ol in the MHD hydrolate was higher, while several compounds were detected in relatively high proportions only in the HD hydrolate. MHD also resulted in a higher phenolic content of the wastewater, where an increase in the concentration of chlorogenic acid was also observed. It can be concluded that the isolation method had a great influence on the profile of sea fennel EOs, especially on their corresponding hydrolates and residual wastewater extracts. Due to their valuable chemical composition, these by-products can be a cost-effective source of bioactive compounds that have great potential for use in various industries.

## 1. Introduction

Essential oils (EOs) are isolates of volatile metabolites, usually obtained by the distillation of different plant materials, and have been used since ancient times for food preservation, in traditional medicine, as pharmaceuticals, and for cosmetic purposes. In general, the methods for isolating EOs can be divided into two groups: (i) conventional, traditional methods (e.g., hydro-distillation (HD), steam distillation, cold pressing, etc.), and (ii) modern, innovative techniques (e.g., microwave-assisted hydro-distillation (MHD), microwave hydro-diffusion and gravity, supercritical fluid extraction, etc.) [1,2].

Of all the EOs isolation methods, hydro-distillation is the most commonly used due to numerous advantages over the others. One of the most important advantages is the use of water as a green extraction solvent as it is environmentally friendly, inexpensive, non-toxic, and non-flammable [3]. HD is also a very simple and low-cost technique. However, it is time-consuming and requires high energy consumption. In addition, elevated temperatures and longer extraction time may lead to losses or chemical modifications of EO components. In order to isolate EOs with higher quality and to optimize the time, yield, and cost of production, new EO extraction methods have been developed in recent decades, which comply with the principles of environmentally friendly process engineering. One of these methods is MHD, which has attracted particular attention due to its unique heating mechanism. In HD, heat transfer within the samples can only be achieved by convection and conduction, while MHD, additionally, utilizes radiation, which significantly accelerates the process [4]. MHD is more expensive due to equipment costs, but the extraction time and energy consumption are lower, and the yield of the final product is, usually, higher. This technique also minimizes the environmental impact because it is solvent-free and is characterized by lower CO_2_ emissions. Therefore, the MHD technique is more suitable for laboratory-scale studies [2,3,4,5,6].

The two distillation processes mentioned above for the isolation of essential oils reduce to four main products: (i) essential oil as the main product and by-products such as (ii) hydrolates (often called hydrosol, floral, or aromatic water), (iii) distillation residual wastewater, and (iv) residual plant material (pomace) [7,8,9,10], the first three being liquid. Hydrolate is the water fraction that remains after the separation of essential oils and, usually, contains some essential oil components and other water-soluble volatile plant constituents (mainly oxygenated compounds with significant biological properties). Recently, hydrolates have enjoyed great popularity in cosmetology, in the food sector as flavouring agents, in traditional medicine/pharmacy, and in aromatherapy [7,8,11,12]. In both HD and MHD, the plant material is in direct contact (immersed) with water, leaving residual wastewater at the end of the process. It contains various water-soluble components, the most important of which are phenolics [7,13,14].

Sea fennel (*Crithmum maritimum* L., Apiaceae) is a wild-growing halophytic plant with remarkable aromatic notes and pleasant sensory properties, which is why it is widely used in food, pharmaceutical, and cosmetic preparations in most Mediterranean countries [15,16]. In recent years, the number of studies on sea fennel has increased exponentially, but there are few studies dealing with the characterization of the by-products of EO isolation. The composition of the residual wastewater that remains after the isolation of sea fennel EO has been studied only in the work of Alves-Silva et al. [13], while, to the authors’ knowledge, there is no research on sea fennel hydrolates and the chemistry of sea fennel EOs and the by-products isolated by the different hydro-distillation methods. Therefore, the aim of this study was to investigate the effects of the isolation method used (HD and MHD) on the chemical profile of sea fennel EOs, hydrolates, and residual isolation wastewaters.

## 2. Results and Discussion

EOs are liquid plant isolates containing complex mixtures of volatile and highly aromatic compounds. Traditionally, they are obtained by distillation from plant material by exposing the plants to steam generated in situ or ex situ. In this process, EO compounds are released from the plant material by vaporization, the steam volatilizes the oil, and the vapour mixture of EO and water condenses. The recovered oil is separated from the water due to the difference in density [17]. As the isolated oil remains in contact with water for some time, various polar hydrophilic volatile compounds that can form hydrogen bonds with water are distributed in the water phase, which is then called hydrolate. The content of volatile aromatic compounds from EOs in hydrolate is usually less than 1 g/L, and they mainly contain oxygenated compounds due to their relatively higher solubility in water compared to the solubility of hydrocarbons. Therefore, hydrolates are also characterized by specific flavours and organoleptic properties [8,11,18]. Another by-product is the non-distilled aqueous phase or wastewater. This fraction is also valuable as it contains large amounts of valuable secondary plant compounds, mainly water-soluble polar phenolics [13,14].

According to Ferhat et al. [5], there are two main MHD mechanisms involved in EO isolation. The first mechanism is based on the selective heating of water in the sample matrix, where microwaves interact with free water molecules distributed in vascular systems and plant glands, which then expand dramatically. This leads to rupture of the tissue and extraction of the oil components into the water phase. The second mechanism is related to the dipolar moments of the EO organic compounds that are able to absorb microwave energy. The compounds with a high dipolar moment are easier to extract because they interact more strongly with microwaves than those with a low dipolar moment.

### 2.1. Essential Oils

The results of the chemical composition of the essential oils of sea fennel isolated by the two distillation methods are summarized in Table 1, while the corresponding chromatograms are shown in Figure 1. The constituents of EO were identified and quantified by gas chromatography coupled with mass spectrometry (GC-MS) (Table 1, Figure 1).

As can be seen from the results, 15 detected compounds differed in their occurrence in the EO samples obtained by HD and MHD. Similar to our previous reports [14,19], as well as other reports on Croatian sea fennel [20], the dominant components were non-polar monoterpene compounds, of which limonene (51.4–53.1%) and sabinene (25.2–27.8%) were found in the highest amounts. In the MHD sample, both compounds were detected in higher amounts. Of the other non-oxygenated compounds, only the content of *p*-cymene was highest in the MHD sample (1.84% in HD and 2.53% in MHD, respectively), while all other compounds were detected in lower amounts in EO obtained by MHD. 

A particularly significant decrease was observed for the content of *α*-pinene (from 4.26 to 2.33%), (*E*)-*β*-ocimene (from 4.04 to 2.96%), *β*-pinene (from 1.77 to 1.14%), and *α*-terpinene (from 1.12 to 0.72%). The yield of oxygenated monoterpenes was significantly lower in both samples, 1.90% in HD EO and 2.09% in MHD EO, with only four compounds from this chemical group detected, namely, (*Z*)-sabinene hydrate, linalool, (*E*)-*p*-mentha-2,8-dien-1-ol, and terpinen-4-ol. Of them, terpinen-4-ol (1.3–1.76%) was found in the highest amount. In Croatian sea fennel, non-oxygenated compounds dominated, accounting for about 91% of the total EO compounds (for both hydrodistillation methods), thus no significant differences in oil profiles were expected due to this specific chemical profile.

Published studies have come to conflicting conclusions regarding the influence of the EO isolation method on the chemical profile of EO. While some of them report significant differences between samples, especially in terms of EO yield [2,5], others confirm that EOs are not significantly affected by the extraction method [4,21]. However, in general, a higher content of oxygenated compounds and a lower number of hydrocarbons are reported in EOs isolated by the MHD method. This can be explained by the higher dipole moment of the oxygenated compounds, which is why they interact more strongly with microwaves and are, therefore, more easily extracted than the monoterpene hydrocarbons with weak dipole moment [4]. Agreement with these observations was found in studies by Lucchesi et al. [22] on basil, garden mint, and thyme; Ferhat et al. [5] on citrus peel’s EO; Filly et al. [6], Moradi et al. [23], and Elyemni et al. [4] on rosemary; and Bui Phuc et al. [2] on mint EO. Cardoso-Ugarte et al. [21] studied the influence of the method used and different extraction parameters on the chemistry of basil essential oil and reported a greater number of detected compounds in the oil obtained by MHD, as well as Ferhart et al. [5] in the citrus peel oils obtained by MHD compared to HD and cold pressing. Finally, almost all the mentioned studies reported significant advantages of MHD distillation over conventional hydrodistillation, especially in terms of short duration and lower energy consumption.

### 2.2. Hydrolates

According to Jensch and Strube [10], about 30% of the EO constituents remain in the hydrolate during hydro-distillation, which represents a major product loss. The main volatile components in hydrolates are monoterpene alcohols, aldehydes, ketones, and sesquiterpene alcohols, while lipophilic compounds such as non-oxygenated monoterpenes are usually absent or present in negligible amounts due to their low water solubility [18]. According to Aćimović et al. [18], the similarity between the composition of EOs and their corresponding hydrolates depends mainly on the relation of hydrocarbons and oxygenated compounds in an EO. When oxygenated compounds dominate in EOs, there is a high similarity between the composition of EOs and hydrolates, while the chemistry of hydrolates differs significantly from the composition of EOs in the case of hydrocarbons.

The results of the chemical composition of sea fennel hydrolates extracted by headspace-solid phase microextraction (HS-SPME) and detected by GC-MS are shown in Table 2 and in Figure 2.

Following the previously stated conclusions from the study by Aćimović et al. [18], the chemistry of sea fennel hydrolates differed significantly from the composition of EOs, as expected. Of the 51 compounds detected in both samples (isolated by HD and MHD), only a few were non-oxygenated and present in relatively low amounts (4.64 and 1.23%, respectively). The content of limonene in the hydrolate HD was almost twice that in the MHD sample (1.16 and 0.61%, respectively). γ-Terpinene was found in trace amounts in both samples, while the presence of α-ionene (0.5%) and longipinene (1.76%) was confirmed only in HD hydrolate.

Terpinen-4-ol was the predominant compound in both HD and MHD samples, with yields of 13.86 and 17.45%, respectively, Figure 2. HD hydrolate was characterised by the presence of (*Z*)-*β*-damascenone (4.80%), 10-(acetylmethyl)-3-carene (13.45%), and (*E*)-*α*-ionone (10.04%), the presence of which was either not confirmed or detected in trace amounts in the MHD sample. The contents of benzacetaldehyde (5.34%) and linalool (1.29%) were significantly higher in the HD than in the MHD sample.

Among the other compounds, the most significant increase in yield in the MHD sample compared to the HD sample was observed for spathulenol (5.8-fold higher amount) and *ar*-tumerol (4.16-fold higher amount). The yields of (*E*)-carveol, isocarveol, and α-terpineol were also higher in the MHD hydrolates with proportions of 4.92 and 5.70%, 1.23 and 4.17%, and 2.92 and 3.85%, respectively. Other dominant components such as (*Z*)-sabinene hydrate, (*E*)-sabinene hydrate, (*E*)-*p*-mentha-2,8-dienol, (*Z*)-*p*-mentha-2,8-dienol, and carvacrol were found in about twice the amounts in the MHD hydrolate. It is interesting that among the compounds detected in the hydrolates there was also dillapiol, which has not been found so far in Croatian sea fennel [14,19,20]. This compound was interesting because, according to Pateira et al. [24], two different chemotypes of sea fennel can be distinguished in terms of its content, and, according to their report, Croatian sea fennel belongs to the chemotype II, with a content of 0–6% dillapiol. In this study, the content of dillapiol in hydrolates was also quite low, 1.12% in HD hydrolate and 1.56% in MHD hydrolate, while its presence in EOs was again not confirmed.

### 2.3. Residual Wastewater

This study also aimed to investigate the phenolic compounds present in the residual wastewater after the isolation of EO, and the results are shown in Table 3 and in Figure 3.

In their study, Irakli et al. [25] pointed out the lack of information on the comprehensive profiling of the residual biomass left after the production of EOs and the importance of utilising their bioactive compounds and further converting them into value-added products that can, potentially, be used in various industries. Our previous study reported on the phenolic compounds detected in the residual water left after EO extraction by conventional hydro-distillation of various parts of the sea fennel plant [14]. Similar to this study, chlorogenic acid was the dominant compound with a concentration of 13.67 mg/g dry extract in the HD sample and 22.18 mg/g in the MHD sample. In Figure 4. the chemical structures of the dominant compounds in all investigated samples are presented.

The presence of this bioactive phenolic compound in sea fennel samples has also been reported by other authors [13,14,26,27,28,29]. As expected, the other two compounds present in the samples at high concentrations were cryptochlorogenic acid and neochlorogenic acid. However, these two compounds were found in lower concentrations in the MHD residual wastewater compared to the samples from HD, as were the levels of *p*-hydroxybenzoic acid and sinapic acid, while the content of ferulic acid was higher in the MHD wastewater extract. This is in agreement with the conclusions of Sánchez-Faure et al. [30] and Siracusa et al. [26], who reported that chlorogenic acid and its isomers are almost the only class of phenolic compounds in sea fennel. Of the flavonoids, only rutin was detected, and its concentration was not significantly different between the HD and MHD samples. In general, the sum of extracted identified phenolics were higher in the MHD sample than in HD, 33.11 and 30.87 mg/g, respectively, also indicating that this novel technique produces by-products with a higher amount of useful phenolic phytochemicals.

## 3. Materials and Methods

### 3.1. Plant Material

Sea fennel (*Crithmum maritimum* L.) aerial parts, in the stage of plant full flowering, were collected in late August 2022 on the coast of the island of Čiovo (43.493389° N, 16.272505° E, Central Dalmatia, Croatia). The identity of the plant species was established by the Ph.D. Tonka Ninčević Runjić (Department of Plant Science, Institute for Adriatic Crops and Karst Reclamatio) using identification keys [31] and assigned to the species *C. maritimum* L. based on their morphology and geographical distribution [32]. Voucher specimens of the plant material were deposited in the herbarium of the Department of Biochemistry, Faculty of Chemistry and Technology, University of Split. Air-dried plant material (two weeks at room temperature in a shaded and aerated place) was used for the extraction of essential oils.

### 3.2. Isolation of EOs, and Preparation of Residual Wastewater Extracts

The essential oils of sea fennel were extracted from the plant material (100 g) by hydro-distillation (HD) and microwave-assisted hydro-distillation (MHD). Hydro-distillation was performed in a Clevenger-type apparatus according to the procedure described in detail by Kulišić-Bilušić et al. [20], while microwave-assisted hydro-distillation (MHD) was performed according to the procedure described by Blažević et al. [33]. In both cases, hydro-distillation was performed using a solvent trap (pentane/diethyl ether = 2/1, *v*/*v*) to prevent the migration of lipophilic components into the water layer. The isolation parameters for both methods are summarised in Table 4. The isolated essential oils were dried over anhydrous sodium sulphate and stored at −20 °C until analysis.

The hydrolates, the water layer under the trap, were also separated and stored at −20 °C until analysis. The wastewater that remained after oil isolation was cooled, filtered, and freeze-dried, and the crude extracts obtained were dissolved in water (at a concentration of 1 mg/mL) and used for further analysis. 

### 3.3. Isolation of Volatile Organic Compounds from Hydrolates by HS-SPME

Solid-phase microextraction (SPME) fibre with DVB/CAR/PDMS coating, 50/30 µm, was purchased from Supelco (Sigma Aldrich, Bellefonte, PA, USA). The fibre used for sample extraction was conditioned at 250 °C for 1 h before use in the unit for conditioning. Hydrolate and NaCl were added to 20 mL glass vials, which were sealed with the magnetic cap and silicone septum. The SPME fibre was inserted into the headspace of the vial, and the samples were thermostatted at 40 °C for 30 min. The fibre was then inserted into the GC injection port for thermal desorption for 3 min. Each sample was analysed in duplicate, and the presented results are mean value ± standard deviation.

### 3.4. Identification of the EOs and Hydrolate Constituents by GC-MS

Identification and quantification of the chemical constituents of the essential oil and hydrolates of sea fennel were performed by GC-MS using a gas chromatograph (gas chromatograph model 8890 GC equipped with an automatic liquid injector model 7693A, and a tandem mass spectrometer (MS) model 7000D GC/TQ (Agilent Inc., Santa Clara, CA, USA). Chromatographic separation was performed on the nonpolar HP-5MS UI column (30 m × 0.25 mm × 0.25 µm, Agilent Inc., Santa Clara, CA, USA). Helium was used as the carrier gas at a flow rate of 1.0 mL min, and the sample injection volume was 1 µL. Analyses were performed using MS full scan (33–350 m/z). The ion source temperature was set at 200 °C, the interface temperature at 250 °C, and the ion voltage at 70 eV. For EOs analysis, the column temperature program was set for the first 3 min at 60 °C and then heated to 246 °C at 3 °C/min and maintained for 25 min isothermally. For hydrolate analyses, the initial oven temperature started with a 3 min isotherm at 40 °C, followed by a temperature increase to 80 °C at a rate of 3 °C/min, followed by another temperature increase at a rate of 10 °C/min to 220 °C, where it was maintained for 5 min. 

Compounds were identified by comparing their retention indices with the series of *n*-hydrocarbons (C_7_–C_30_, Supelco Inc., Sigma Aldrich) analysed under the same conditions as described above for both groups of samples, EOs, and hydrolates, respectively. The identification of each component was based on a software comparison of mass spectra with library entries from two commercial databases, Wiley 7 MS library (Wiley, NY, USA) and NIST02 (Gaithersburg, MD, USA), as well as the comparison of their mass spectra and linear retention indices with published data [34]. Relative percentages of the oil components (%) were calculated based on the peak areas on HP-5MS UI column and are reported in tables according to their elution order, while Linear Retention Indexes (LRIs) were computed using van den Dool and Kratz’s equation [35]. Analyses were performed in duplicate, and the percentages shown in Table 2 and Table 3 were calculated as the mean value ± standard deviation.

### 3.5. Identification of the Wastewater Chemical Components by HPLC

The wastewater extracts were dissolved in water (1 mg/mL) and injected into a high-performance liquid chromatograph (Shimadzu Nexera LC-40, Shimadzu, Kyoto, Japan) with a UV-Vis detector. Separation was performed at 35 °C using a Phenomenex C18 reversed-phase column (250 mm × 4.6 mm, 5 μm; Torrance, CA, USA) with a binary gradient mixture of solvent A (ultrapure water/85% *o*-phosphoric acid = 99.8/0.2, *v*/*v*) and solvent B (methanol/acetonitrile = 1/1, *v*/*v*) at a flow rate of 1 mL/min: 0–16 min, 4–15% B; 16–50 min, 15–35% B; 50–62 min, 35–4% B; 65 min, 4% B. Phenolic compounds were identified by comparing retention times and absorption maxima at 220 and 320 nm with data obtained for corresponding standards and by sample spiking. Concentrations were expressed as mg of compound per gram of dry extract (mg/g).

## 4. Conclusions

Nowadays, the interest and number of studies on aromatic and medicinal plants have increased due to the exponentially growing popularity of the use of natural products in various industries. While the application and benefits of various EOs have been extensively studied, this research confirms the great chemical potential of their liquid by-products, which were usually treated as waste (hydrolate and residual wastewater). The isolation process is an essential element to improve the overall quality of EO. Hydro-distillation by long heating (up to several hours) (as in the case of conventional HD), and exposure to microwaves (as in the case of MHD) have a great influence on the profile of sea fennel EOs and especially on their corresponding hydrolates. In addition, the wastewater from MHD had a higher phenolic content compared to HD. Due to the much lower time and energy consumption, MHD is a good alternative for the conventional HD of sea fennel EOs. Additionally, the liquid by-products of EO isolation, both the hydrolates and the wastewater, can be used to obtain high value-added products from low-cost raw materials that are often treated as waste.

## Figures and Tables

**Figure 1 plants-12-01466-f001:**
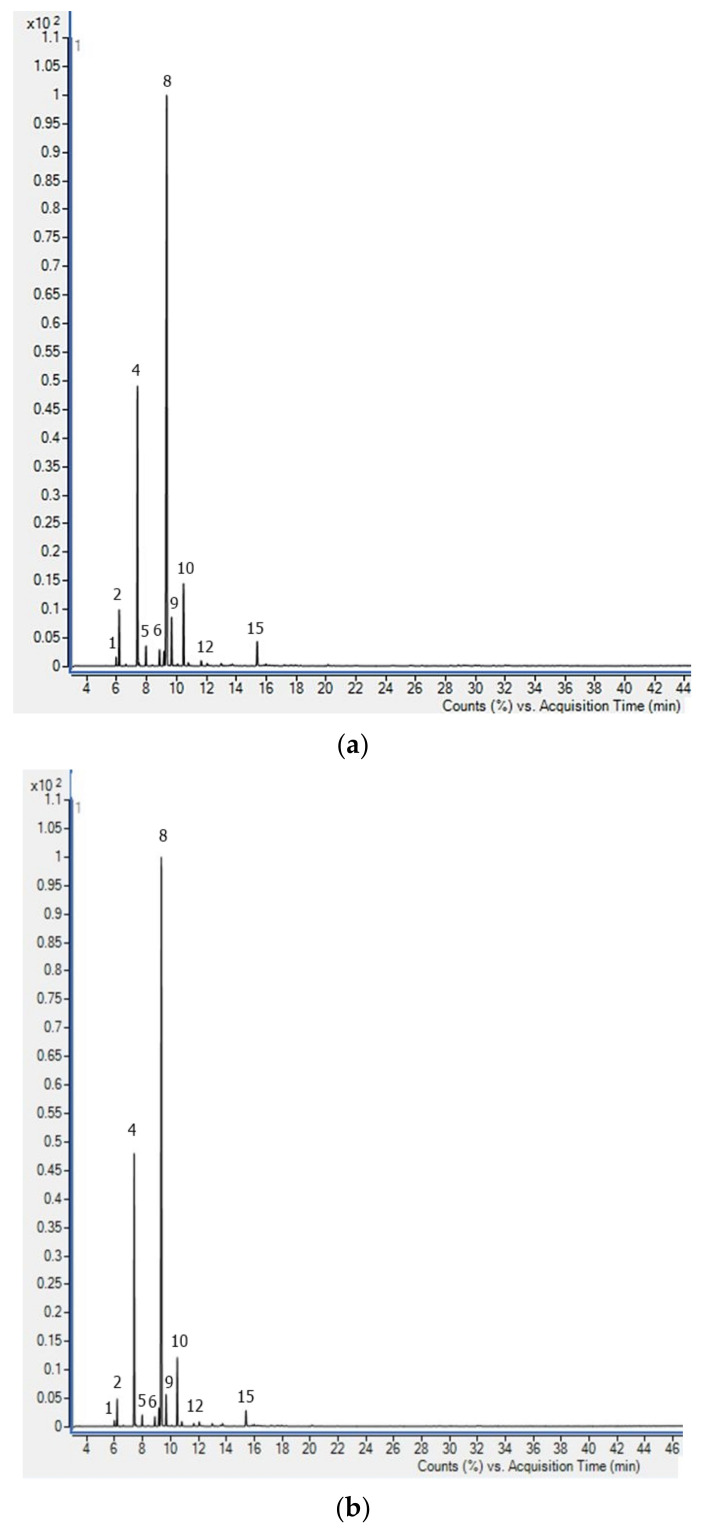
GC-MS total ion chromatogram of sea fennel essential oil obtained hydro-distillation (**a**) and microwave-assisted hydro-distillation (**b**). Peak assignment is given in Table 1.

**Figure 2 plants-12-01466-f002:**
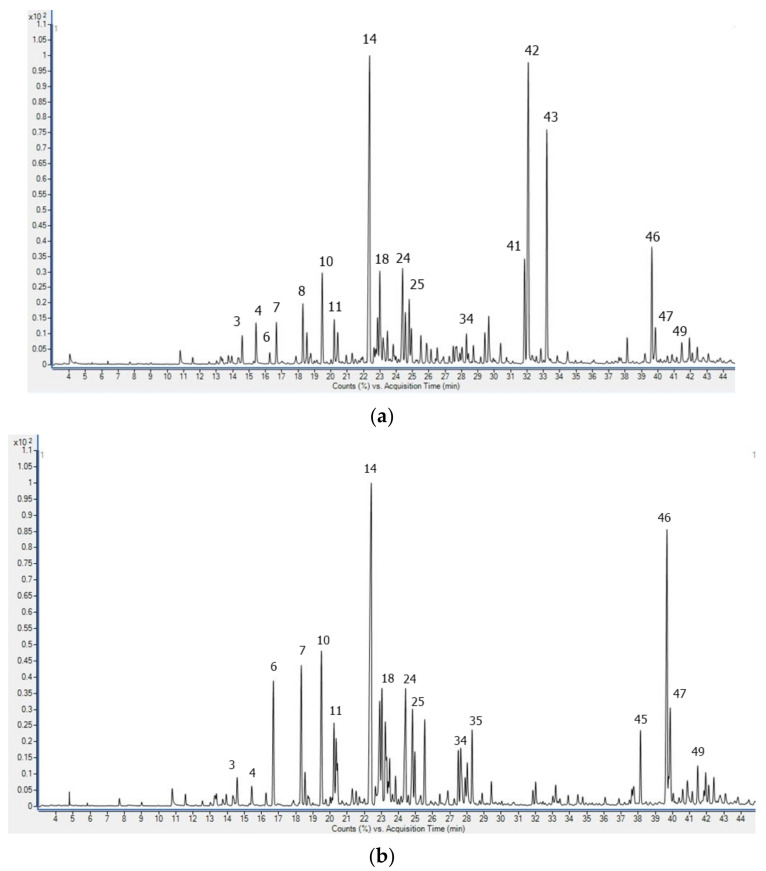
GC-MS total ion chromatogram of sea fennel hydrolates obtained hydro-distillation (**a**) and microwave-assisted hydro-distillation (**b**). Peak assignment is given in Table 2.

**Figure 3 plants-12-01466-f003:**
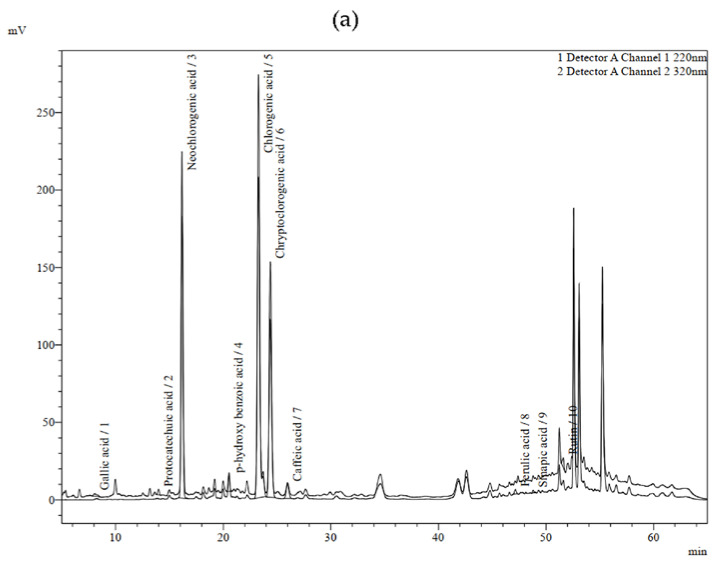
HPLC chromatogram of residual wastewater extracts after essential oil extraction by hydro-distillation (**a**) and microwave-assisted hydro-distillation (**b**). Peak assignment is given in Table 3.

**Figure 4 plants-12-01466-f004:**
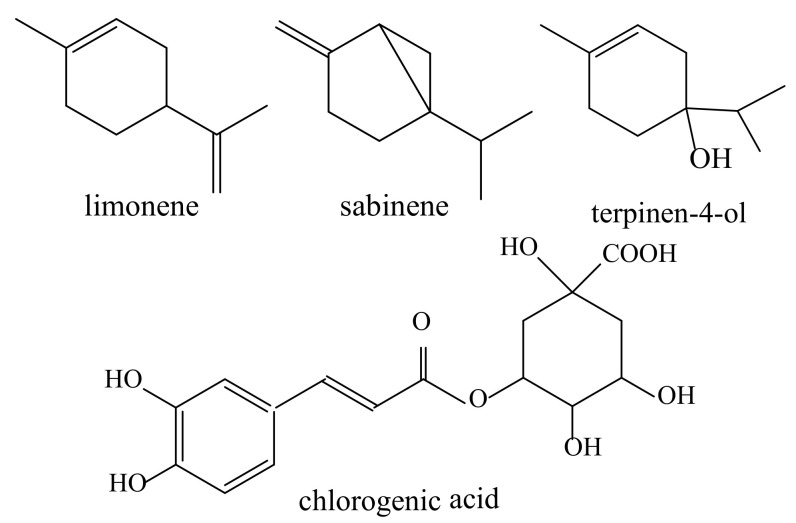
Major components detected in sea fennel essential oils, hydrolates and residual wastewater.

**Table 1 plants-12-01466-t001:** Chemical composition of sea fennel essential oils isolated by hydro-distillation (HD) and microwave-assisted hydro-distillation (MHD), analysed by gas chromatography coupled with mass spectrometry (GC-MS).

No.	Component	Rt	LRI	Literature LRI	HD (%)	MHD (%)	Mode of Identification
1.	*α*-thujene	5.989	926	924	0.69 ± 0.13	0.50 ± 0.09	GC, MS
2.	*α*-pinene	6.195	932	932	4.26 ± 0.84	2.33 ± 0.44	GC, MS
3.	camphene	6.608	946	946	tr	nd	GC, MS
4.	sabinene	7.410	972	969	25.24 ± 2.64	27.81 ± 4.30	GC, MS
5.	*ß*-pinene	7.986	991	974	1.77 ± 0.43	1.14 ± 0.26	GC, MS
6.	*α*-terpinene	8.897	1016	1014	1.12 ± 0.05	0.72 ± 0.09	GC, MS
7.	*p*-cymene	9.175	1023	1020	1.84 ± 0.65	2.53 ± 0.09	GC, MS
8.	limonene	9.337	1027	1024	51.38 ± 4.42	53.05 ± 5.20	GC, MS
9.	(*E*)-*ß*-ocimene	9.698	1037	1044	4.04 ± 0.59	2.96 ± 0.38	GC, MS
10.	*γ*-terpinene	10.506	1057	1054	7.12 ± 0.73	6.54 ± 0.43	GC, MS
11.	(*Z*)-sabinene hydrate	10.820	1065	1065	0.14 ± 0.13	0.41 ± 0.06	GC, MS
12.	terpinolene	11.689	1088	1086	0.38 ± 0.08	0.15 ± 0.03	GC, MS
13.	linalool	12.073	1097	1095	nd	0.38 ± 0.06	GC, MS
14.	(*E*)-*p*-mentha-2,8-dien-1-ol	13.024	1120	-	tr	nd	GC, MS
15.	terpinen-4-ol	15.438	1177	1174	1.76 ± 0.04	1.30 ± 0.31	GC, MS
*Total identified (%)*					*99.82*	*99.74*	
*Total non-oxygenated compounds* *Total oxygenated compounds*					*97.92* *1.90*	*97.73* *2.09*	

Rt—Retention time, LRI—Linear Retention Index, tr—traces (<0.10%), nd—not detected.

**Table 2 plants-12-01466-t002:** HS-SPME-GC-MS analysis of sea fennel hydrolates obtained by hydro-distillation (HD) and microwave-assisted hydro-distillation (MHD).

No.	Component	Rt	LRI	Literature LRI	HD (%)	MHD (%)	Mode of Identification
1.	benzaldehyde	10.747	950	952	0.46 ± 0.11	0.70 ± 0.02	GC, MS
2.	(*E*,*E*)-heptadienal	13.695	1005	1005	0.91 ± 0.04	tr	GC, MS
3.	limonene	14.552	1021	1024	1.16 ± 0.11	0.61 ± 0.02	GC, MS
4.	benzacetaldehyde	15.402	1037	1036	5.34 ± 0.01	1.15 ± 0.04	GC, MS
5.	γ-terpinene	16.259	1054	1054	tr	tr	GC, MS
6.	(*Z*)-sabinene hydrate	16.659	1061	1065	1.54 ± 0.26	3.95 ± 0.04	GC, MS
7.	(*E*)-sabinene hydrate	18.294	1093	1098	2.37 ± 0.36	4.72 ± 0.08	GC, MS
8.	linalool	18.536	1097	1095	1.29 ± 0.04	0.60 ± 0.01	GC, MS
9.	6-methyl-3,5-heptadien-2-one	18.750	1101	1107	0.45 ± 0.05	tr	GC, MS
10.	(*E*)-*p*-mentha-2,8-dienol	19.488	1116	1118	2.40 ± 0.03	4.34 ± 0.83	GC, MS
11.	(*Z*)-*p*-mentha-2,8-dienol	20.216	1131	1133	1.49 ± 0.17	2.50 ± 0.13	GC, MS
12.	(*E*)-limonene oxide	20.359	1134	1137	nd	1.84 ± 0.37	GC, MS
13.	(*E*)-*p*-menth-2-en-1-ol	20.421	1135	1136	1.38 ± 0.21	1.18 ± 0.07	GC, MS
14.	terpinen-4-ol	22.446	1176	1174	13.86 ± 0.56	17.45 ± 0.87	GC, MS
15.	3-methylacetophenone	22.649	1180	1179	1.13 ± 0.02	0.74 ± 0.20	GC, MS
16.	*p*-cymen-8-ol	22.794	1183	1179	0.82 ± 0.05	tr	GC, MS
17.	isocarveol	22.896	1185	1187	1.23 ± 0.06	4.17 ± 0.32	GC, MS
18.	*α*-terpineol	23.032	1188	1186	2.92 ± 0.02	3.85 ± 0.03	GC, MS
19.	isocarvomenthone	23.078	1189	1194	0.87 ± 0.01	Tr	GC, MS
20.	(*Z*)-carvomenthone	23.221	1192	1194	nd	2.52 ± 0.01	GC, MS
21.	(*Z*)-dihydrocarvone	23.307	1194	-	nd	1.12 ± 0.12	GC, MS
22.	(*E*)-carvomenthone	23.482	1197	-	1.53 ± 0.10	1.63 ± 0.40	GC, MS
23.	(*E*)-piperitol	23.826	1204	1207	0.35 ± 0.02	0.45 ± 0.03	GC, MS
24.	(*E*)-carveol	24.438	1218	1215	4.92 ± 0.29	5.70 ± 0.35	GC, MS
25.	neoiso dihydro carveol	24.591	1221	1226	3.03 ± 0.16	3.20 ± 0.46	GC, MS
26.	(*Z)*-carveol	24.828	1226	1226	1.58 ± 0.01	1.25 ± 0.35	GC, MS
27.	carvone	25.520	1241	1239	1.10 ± 0.03	tr	GC, MS
28.	α-ionene	25.848	1248	-	0.50 ± 0.01	nd	GC, MS
29.	(*E*)-2-decanal	26.424	1261	1260	nd	tr	GC, MS
30.	perillaldehyde	26.800	1269	-	nd	tr	GC, MS
31.	bornyl acetate	27.492	1284	1283	0.38 ± 0.01	2.04 ± 0.25	GC, MS
32.	*p*-cymen-7-ol	27.631	1287	1287	0.68 ± 0.04	3.54 ± 0.87	GC, MS
33.	thymol	27.899	1293	1289	0.38 ± 0.02	0.97 ± 0.04	GC, MS
34.	carvacrol	28.306	1301	1298	2.62 ± 0.02	4.32 ± 0.02	GC, MS
35.	*p*-vinyl guaiacol	28.719	1311	1309	1.63 ± 0.25	nd	GC, MS
36.	(*E*,*E*)-2,4-decadienal	28.879	1315	1315	nd	0.47 ± 0.01	GC, MS
37.	myrtenyl acetate	29.431	1327	1324	1.63 ± 0.13	0.49 ± 0.03	GC, MS
38.	longipinene	29.666	1333	1350	1.76 ± 0.06	nd	GC, MS
39.	1H-indene,2,3-dihydro-1,1,4,5-tetramethyl	30.383	1349	-	1.22 ± 0.03	nd	GC, MS
40.	(*Z*)-*β*-damascenone	31.859	1383	1387	4.80 ± 0.34	tr	GC, MS
41.	10-(acetylmethyl)-3-carene	32.121	1390	-	13.45 ± 0.61	nd	GC, MS
42.	*α*-ionone	33.245	1416	1428	10.04 ± 0.33	tr	GC, MS
43.	*α*-copaen-11-ol	37.739	1527	1539	nd	0.52 ± 0.10	GC, MS
44.	cyperene epoxyde	38.136	1537	-	tr	1.99 ± 0.11	GC, MS
45.	spathulenol	39.648	1575	1577	1.50 ± 0.12	8.78 ± 0.37	GC, MS
46.	*ar*-tumerol	39.767	1579	1582	0.88 ± 0.06	3.66 ± 0.89	GC, MS
47.	*β*-copaen-4-α-ol	39.870	1581	1590	tr	1.17 ± 0.13	GC, MS
48.	dill apiol	41.474	1624	1620	1.12 ± 0.12	1.56 ± 0.12	GC, MS
49.	muurola-4,10(14)-dien-1*β*-ol	41.941	1636	1630	tr	0.56 ± 0.00	GC, MS
50.	4,5-dehydro isolongifolene	42.415	1649	-	tr	0.62 ± 0.04	GC, MS
*Total identified (%)*					*94.72*	*94.36*	
*Total non-oxygenated compounds* *Total oxygenated compounds*					*4.64* *90.08*	*1.23* *93.13*	

Rt—Retention time, LRI—Linear Retention Index, tr—traces (<0.10%), nd—not detected.

**Table 3 plants-12-01466-t003:** Phenolic profile (mg/g of dry extract) of residual wastewater extracts after essential oil extraction by hydro-distillation (HD) and microwave-assisted hydro-distillation (MHD) detected by high performance liquid chromatography (HPLC).

No.	Component	HD	MHD
1.	gallic acid	0.03 ± 0.01	0.02 ± 0.00
2.	protocatechuic acid	0.05 ± 0.02	0.04 ± 0.00
3	neochlorogenic acid	7.92 ± 0.04	4.48 ± 0.11
4.	*p*-hydroxybenzoic acid	0.78 ± 0.06	0.53 ± 0.02
5.	chlorogenic acid	13.67 ± 0.03	22.18 ± 0.03
6.	cryptochlorogenic acid	7.75 ± 0.05	5.17 ± 0.02
7.	caffeic acid	0.04 ± 0.01	0.03 ± 0.02
8.	ferulic acid	0.11 ± 0.00	0.22 ± 0.00
9.	sinapic acid	0.16 ± 0.01	0.11 ± 0.00
10.	rutin	0.35 ± 0.00	0.34 ± 0.01

**Table 4 plants-12-01466-t004:** Parameters of the used essential oil isolation methods.

	Hydro-Distillation	Microwave-Assisted Hydro-Distillation
**Equipment**	Clevenger-type apparatus	MAE Ethos X, (Milestone Srl, Italy)
**Heating time ***	15 min	3 min
**Temperature**	100 °C	98 °C
**Duration**	3 h	30 min
**Soaking**	Yes	Yes
**Other**		Microwave power 500 Hz

* time needed to obtain the distillation of the essential oil first droplet.

## Data Availability

Not applicable.

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
