# Peer review of "Conventional vs. Microwave-Assisted Hydrodistillation: Influence on the Chemistry of Sea Fennel Essential Oil and Its By-Products"

_plants, 2023, doi:10.3390/plants12071466_

Round 1

Reviewer 1 Report

This paper describes two procedures for obtaining chemical profiles of sea fennel. The isolation of essential oils has already been established in other works and the results are in general agreement with those obtained by other authors.

The most interesting part of the work is related to the characterization of the main constituents present in the hydrolate and in the residual wastewater obtained in the extraction process of the essential oils.

The approach of the work is adequate, as well as the development of the analytical techniques and procedures used for the identification of compounds, treatment of the samples, and analysis of the results (although some comments will be made later regarding the quantification process of the identified compounds). Therefore, from my point of view, the work deserves to be accepted for publication after major revision.

General comments

The quantification procedure is the main limitation of the work. It is surprising that the authors exclusively use the MS full scan mode to identify the compounds as well as to quantify them. It is well known that the recording of mass spectra in SIM mode (recording of selected ions) considerably increases the signal-to-noise ratio (S/N) and significantly improves the sensitivity of the procedure. In this sense, it would be advisable for the authors to consider this possibility in order to achieve not only the identification of some other minority compounds in the simples, but also their possible quantification.

The liquid by-products of the isolation of essential oils are proposed in the work to obtain high value-added products from low-cost materials. However, in order to adequately assess this aspect, it would be necessary to determine the absolute contents of these compounds, especially in the fraction corresponding to sea fennel hydrolates.

Specific comments

In the text the terms MHD and MAE are used interchangeably to refer to the same microwave extraction procedure. The terminology should be homogenized.

Lines 309-312 (tables 2 and 3)

Results are expressed as mean value of 2 determinations. It would be interesting to know both values or, failing that, their corresponding standard deviation in order to evaluate more adequately the significance level of the results.

On the other hand, it is specified that no correction factors are used. It is not clear which correction factors are the authors referring to. Pleased, specify them.

Author Response

Reviewer #1

This paper describes two procedures for obtaining chemical profiles of sea fennel. The isolation of essential oils has already been established in other works and the results are in general agreement with those obtained by other authors. The most interesting part of the work is related to the characterization of the main constituents present in the hydrolate and in the residual wastewater obtained in the extraction process of the essential oils. The approach of the work is adequate, as well as the development of the analytical techniques and procedures used for the identification of compounds, treatment of the samples, and analysis of the results (although some comments will be made later regarding the quantification process of the identified compounds). Therefore, from my point of view, the work deserves to be accepted for publication after major revision.

Response: We would like to thank the Reviewer for the positive comments and all suggestions that helped us improve the manuscript.

General comments

The quantification procedure is the main limitation of the work. It is surprising that the authors exclusively use the MS full scan mode to identify the compounds as well as to quantify them. It is well known that the recording of mass spectra in SIM mode (recording of selected ions) considerably increases the signal-to-noise ratio (S/N) and significantly improves the sensitivity of the procedure. In this sense, it would be advisable for the authors to consider this possibility in order to achieve not only the identification of some other minority compounds in the simples, but also their possible quantification.

Response: Thank you for your comment, however, in this research the quantification procedure was not applied, since we aimed to identify and determine the presence of compounds in the analysed samples. GC-MS analysis (MS full scan mode) was used exclusively for the qualitative determination of the presence of compounds, i.e. the percentages in which they are present in the samples were determined, for the purpose of their mutual comparison.

The liquid by-products of the isolation of essential oils are proposed in the work to obtain high value-added products from low-cost materials. However, in order to adequately assess this aspect, it would be necessary to determine the absolute contents of these compounds, especially in the fraction corresponding to sea fennel hydrolates.

Response: We would like to thank the reviewer for the careful observation and express our agreement with his comment. However, the quantification of volatiles (key compounds) in hydrolates is questionable due to used method of their isolation (SPME, solid instead of liquid extraction) and detection (GC-MS; volatiles from hydrophilic/non-volatile phase), so published papers are focused only on their qualitative composition like in present study. Also, in this study the amounts (volumes) of hydrolates in HD and MHD are different what additional complicate the comparison of the results.

 Specific comments

In the text the terms MHD and MAE are used interchangeably to refer to the same microwave extraction procedure. The terminology should be homogenized.

 Response: Corrected.

Lines 309-312 (tables 2 and 3)

Results are expressed as mean value of 2 determinations. It would be interesting to know both values or, failing that, their corresponding standard deviation in order to evaluate more adequately the significance level of the results.

Response: Following the comment, we icluded standard deviation in Tables.

On the other hand, it is specified that no correction factors are used. It is not clear which correction factors are the authors referring to. Pleased, specify them.

Response: We would like to thank the reviewer for this remark esspecially since this was writing mistak. The sentance has been removed from the 

Reviewer 2 Report

This manuscript entitled “Conventional vs. Microwave-assisted hydrodistillation: Influence on the chemistry of sea fennel essential oil and its by products” by Politeo et al., describes results on chemical components identification of sea fennel essential oils extracted using conventional and microwave-assisted techniques. The plant essential oils have already been studied by several authors earlier but the investigation of this plant using different extraction techniques and chemical components identification of hydrolates and waste water might be valuable for the scientist working in the similar field. However, there are several issues with this manuscript which must be taken care of before the manuscript is accepted for its publication.

- Identification of essential oil compounds only on single column always lead to wrong characterization of volatile components. Therefore, Authors should provide the analysis of essential oil composition on two different columns such as polar and nonpolar columns in order to achieve authentic identification of more oil components

- Biological activity evaluation and their comparison would be an interesting part of the MS why authors did not studied the biological activity of the extracts and essential oils?.

-In Table-1 and Table-2, KI should be replaced with LRI

-GC chromatograms of the essential oils analysis indicating peaks of all the identified compounds must be included in the manuscript.

-In order to improve the identification of volatile oils components and improve the quality of the manuscript, I would like encourage the authors to include  the following references and cite them in the MS

1. Compositional characteristics of the essential oil of Myrtus communis grown in the central part of Saudi Arabia

Journal of Essential Oil Research 26 (1), 13-18, 2014

2. A detailed study on chemical characterization of essential oil components of two Plectranthus species grown in Saudi Arabia

Journal of Saudi Chemical Society 20 (6), 711-721, 2016, https://doi.org/10.1016/j.jscs.2016.03.006

3. Chemical composition of vegetative parts and flowers essential oils of wild Anvillea garcinii grown in Saudi Arabia

3. Records of Natural Products 10 (2), 251-256, 2016.

-Authors should provide LRI values of identified components in Table-1 and Table -2 from the literature and aforementioned references could be used and cited in the MS.

-There are so many English grammar and typographical mistakes in the manuscript which must be rectified.

-Authors should provide the calculation methods for LRI determination and how they were calculated.

-Identification methodology of the essential oil components should be provide in the experimental section.

- Authors should provide the detail of the proper identification of plant used in this manuscript. Was the plants were identified by a plant taxonomist/botanist? Please give detail of the plant taxonomist and voucher specimens of the plant.

-provide the chromatograms of the HPLC analysis

Author Response

Reviewer #2

This manuscript entitled “Conventional vs. Microwave-assisted hydrodistillation: Influence on the chemistry of sea fennel essential oil and its by products” by Politeo et al., describes results on chemical components identification of sea fennel essential oils extracted using conventional and microwave-assisted techniques. The plant essential oils have already been studied by several authors earlier but the investigation of this plant using different extraction techniques and chemical components identification of hydrolates and waste water might be valuable for the scientist working in the similar field. However, there are several issues with this manuscript which must be taken care of before the manuscript is accepted for its publication.

Response: We would like to thank the Reviewer for the time, efforts, positive comments and suggestions.

- Identification of essential oil compounds only on single column always lead to wrong characterization of volatile components. Therefore, Authors should provide the analysis of essential oil composition on two different columns such as polar and nonpolar columns in order to achieve authentic identification of more oil components.

Response: Although we partially agree with the reviewer (but we would not say wrong characterization), at this moment we cannot performed the analysis using polar column.

- Biological activity evaluation and their comparison would be an interesting part of the MS why authors did not study the biological activity of the extracts and essential oils?

Response: This study is focused only on the chemical composition of the samples (similar as in papers that reviewer suggested in comment 6) obtained by two different methods, while another paper on their biological activity (antioxidant and antimicrobial) and composition/structure-activity relationship (SAR) is in progress.

  • In Table-1 and Table-2, KI should be replaced with LRI

Response: In Table 1 and Table 2 KI is replaced with LRI.

  • GC chromatograms of the essential oils analysis indicating peaks of all the identified compounds must be included in the manuscript.

Response: GC chromatograms of the essential oils analysis indicating peaks of all the identified compounds were inserted.

-In order to improve the identification of volatile oils components and improve the quality of the manuscript, I would like encourage the authors to include the following references and cite them in the MS

  1. Compositional characteristics of the essential oil of Myrtus communis grown in the central part of Saudi Arabia

Journal of Essential Oil Research 26 (1), 13-18, 2014

  1. A detailed study on chemical characterization of essential oil components of two Plectranthus species grown in Saudi Arabia

Journal of Saudi Chemical Society 20 (6), 711-721, 2016, https://doi.org/10.1016/j.jscs.2016.03.006

  1. Chemical composition of vegetative parts and flowers essential oils of wild Anvillea garcinii grown in Saudi Arabia

Records of Natural Products 10 (2), 251-256, 2016.

Response: According to the Reviewer comment the suggested references have been checked and unfortunately according to the author’s opinion they do not align with the concept of this study.

-Authors should provide LRI values of identified components in Table-1 and Table -2 from the literature and aforementioned references could be used and cited in the MS.

Response: The LRI values from the literature (Adams 2017; Wiley 7 / NIST02) have been inserted.

-There are so many English grammar and typographical mistakes in the manuscript which must be rectified.

Response: The whole manuscript has been carefully checked and revised.

-Authors should provide the calculation methods for LRI determination and how they were calculated.

Response: LRI were determined using van den Dool and Kratz’s equation:

tR- n-alkane standard retention time, z- number of C-atoms

Reference:

  1. H.V.D. Dool, P.D. Kratz, A generalization of the retention index system including linear temperature programmed gas-liquid partition chromatography, J. Chromatogr. A, 11 (1963), pp. 463-471

-Identification methodology of the essential oil components should be provide in the experimental section.

Response: According to the Reviewer suggestion we have improved this issue in the paper.

- Authors should provide the detail of the proper identification of plant used in this manuscript. Was the plants were identified by a plant taxonomist/botanist? Please give detail of the plant taxonomist and voucher specimens of the plant.

Response: The plant material section has been corrected according to the Reviewer comment.

-provide the chromatograms of the HPLC analysis

Response: The chromatograms are now included in MS.

Round 2

Reviewer 1 Report

I still think that recording chromatograms in SCAN/SIM mode would allow not only a better characterization of the analytes but also a better quantification.

I strongly recommend the authors to work in this way in their next publications.

Author Response

Reviewer #1

I still think that recording chromatograms in SCAN/SIM mode would allow not only a better characterization of the analytes but also a better quantification. I strongly recommend the authors to work in this way in their next publications.

Response: We would like again to thank the Reviewer #1 for the suggestion and we will surely take it in account in our next publications.

Reviewer 2 Report

Now authors have improved the manuscript to some extent. However, there are still several issues with this manuscript which must be taken care of before the manuscript is processed further.

- I have already suggested in my previous review, to provide the name of the botanist who has identified the plant amterial but authors did not provide the detail of the botanist such name, institution etc.. Authors must provide the name of the botanist who had identified the plant materials.

- I would like to encourage the authors to cite following references

a. Compositional characteristics of the essential oil of Myrtus communis grown in the central part of Saudi Arabia. Journal of Essential Oil Research 26 (1), 13-18, 2014

b. A detailed study on chemical characterization of essential oil components of two Plectranthus species grown in Saudi Arabia. Journal of Saudi Chemical Society 20 (6), 711-721, 2016, https://doi.org/10.1016/j.jscs.2016.03.006

c. Chemical composition of vegetative parts and flowers essential oils of wild Anvillea garcinii grown in Saudi Arabia. Records of Natural Products 10 (2), 251-256, 2016.

These references are important and very much align with the concept of the research. They are very much suitable in the introduction section for the sentences like (line 46-47) “Of all the EOs isolation methods, hydro-distillation is the most commonly used due to numerous advantages over the others [a-c]”

-Line 46: Replavce “hydro-distilation” with “hydro-distillation”

-authors should provide the detail of n-hydrocarbons used for the calculation of LRI values for example name of n-hydrocarbons used (Cn-Cn), product number, company name they acquired from etc.

-Did authors inject the mixture of n-hydrocarbons for the calculation of LRI values. Was it injected before oil samples or after the oil sample.

-what was the GC method used for the analysis n-hydrocarbon mixtures. It should be clearly mentioned in the experimental section of manuscript.

-How they computed the LRI values? it must be described in experimental section of the manuscript.

-Authors are claiming that they used LRI values from the literatures however the did not cite those literatures in the manuscript why???

-For the LRI values comparisons from the literatures, aforementioned references [a-c] can be cited in the manuscript as in these references, LRI values of volatile components are calculated on HP-5MS column. While in the present study authors have also used the same column for the oil analysis.

-Authors have not mentioned anything regarding the GC-FID analysis of volatile oils. What was the methods for GC-FID analysis.

-Authors should provide the percentage of oil components from GC-FID analysis as they are more reliable and give better results for quantitative analysis.

-Authors should add the biological activity of the essential oils and comparisons of the results of essential oils obtained with HD and MHD techniques. Only chemical composition without biological activity is not enough for a study to be published in journals with high impact factors like “Plants” especially when the plants has already been studied earlier by various scientists.

-Authors have used two different GC methods for the analysis of hydrolats and essential oils why???

-Authors should add a column for the retention times in Tables-1 and Table-2 for more clarity of the essential oil components.

-Why authors did not include content and composition of hydrolats in Tables.

-Authors must include the chromatograms for the hydrolats analysis.

- There are still so many English grammar and typographical mistakes in the manuscript which must be rectified.

Author Response

Reviewer #2

Now authors have improved the manuscript to some extent. However, there are still several issues with this manuscript which must be taken care of before the manuscript is processed further.

- I have already suggested in my previous review, to provide the name of the botanist who has identified the plant amterial but authors did not provide the detail of the botanist such name, institution etc.. Authors must provide the name of the botanist who had identified the plant materials.

Response: Corrected.

- I would like to encourage the authors to cite following references

  1. Compositional characteristics of the essential oil of Myrtus communis grown in the central part of Saudi Arabia. Journal of Essential Oil Research 26 (1), 13-18, 2014
  2. A detailed study on chemical characterization of essential oil components of two Plectranthus species grown in Saudi Arabia. Journal of Saudi Chemical Society 20 (6), 711-721, 2016, https://doi.org/10.1016/j.jscs.2016.03.006
  3. Chemical composition of vegetative parts and flowers essential oils of wild Anvillea garcinii grown in Saudi Arabia. Records of Natural Products 10 (2), 251-256, 2016.

These references are important and very much align with the concept of the research. They are very much suitable in the introduction section for the sentences like (line 46-47) “Of all the EOs isolation methods, hydro-distillation is the most commonly used due to numerous advantages over the others [a-c]”

Response: Unfortunately, we found this papers interesting and well written but still not so important for this study.

-Line 46: Replavce “hydro-distilation” with “hydro-distillation”

Response: Corrected.

-authors should provide the detail of n-hydrocarbons used for the calculation of LRI values for example name of n-hydrocarbons used (Cn-Cn), product number, company name they acquired from etc.

Response: Added.

-Did authors inject the mixture of n-hydrocarbons for the calculation of LRI values. Was it injected before oil samples or after the oil sample.

Response: The mixture of n-hydrocarbons has been injected prior the EOs/hydrolate analysis, as well as at final after the EOs/hydrolate analysis using both methods.

-what was the GC method used for the analysis n-hydrocarbon mixtures. It should be clearly mentioned in the experimental section of manuscript.

Response: Yes. This section has been corrected.

-How they computed the LRI values? it must be described in experimental section of the manuscript.

Response: Added.

-Authors are claiming that they used LRI values from the literatures however the did not cite those literatures in the manuscript why???

Response: It was oversight with only author name written in text. Corrected.

-For the LRI values comparisons from the literatures, aforementioned references [a-c] can be cited in the manuscript as in these references, LRI values of volatile components are calculated on HP-5MS column. While in the present study authors have also used the same column for the oil analysis.

Response: Thank you for this suggestion.

-Authors have not mentioned anything regarding the GC-FID analysis of volatile oils. What was the methods for GC-FID analysis.

Response: This analyses were not performed, only GC-MS as have been stated in paper.

-Authors should provide the percentage of oil components from GC-FID analysis as they are more reliable and give better results for quantitative analysis.

Response: Agree, but this study was focused on identification so GC-FID was not performed.

-Authors should add the biological activity of the essential oils and comparisons of the results of essential oils obtained with HD and MHD techniques. Only chemical composition without biological activity is not enough for a study to be published in journals with high impact factors like “Plants” especially when the plants has already been studied earlier by various scientists.

Response: As we already stated, this parameters are part of the other study which is in progress at the moment. This research focuse only on chemical componentes of the isolates.

-Authors have used two different GC methods for the analysis of hydrolats and essential oils why???

Response: Yes, as diffreent methods for oil samples and HS-SPME (EOs vs. VOCs) has been previously reported in literature.

-Authors should add a column for the retention times in Tables-1 and Table-2 for more clarity of the essential oil components.

Response: Rt are now added in Tables.

-Why authors did not include content and composition of hydrolats in Tables.

Response: They were included in the original version of manucript (Table 2), and still are there.

-Authors must include the chromatograms for the hydrolats analysis.

Response: They have been included in the first revised version (Figure 2), and still are there.

- There are still so many English grammar and typographical mistakes in the manuscript which must be rectified.

Response: Corrected (Instatext).

Round 3

Reviewer 2 Report

Authors have responded to comments and the MS could be accepted for its publication.